# The Potential Role of RANTES in Post-Stroke Therapy

**DOI:** 10.3390/cells12182217

**Published:** 2023-09-06

**Authors:** Hanna Pawluk, Renata Kołodziejska, Grzegorz Grześk, Alina Woźniak, Mariusz Kozakiewicz, Agnieszka Kosinska, Mateusz Pawluk, Magdalena Grześk-Kaczyńska, Elżbieta Grzechowiak, Jakub Wojtasik, Grzegorz Kozera

**Affiliations:** 1Department of Medical Biology and Biochemistry, Faculty of Medicine, Collegium Medicum in Bydgoszcz, Nicolaus Copernicus University in Toruń, Karłowicza 24, 85-092 Bydgoszcz, Poland; alina-wozniak@wp.pl (A.W.); pawluk.mateusz23@gmail.com (M.P.); 2Department of Cardiology and Clinical Pharmacology, Faculty of Health Sciences, Collegium Medicum in Bydgoszcz, Nicolaus Copernicus University in Toruń, Ujejskiego 75, 85-168 Bydgoszcz, Poland; ggrzesk@cm.umk.pl (G.G.); magdalenagrzesk@gmail.com (M.G.-K.); 3Division of Biochemistry and Biogerontology, Department of Geriatrics, Faculty of Health Sciences, Collegium Medicum in Bydgoszcz, Nicolaus Copernicus University in Toruń, Dębowa 3, 85-626 Bydgoszcz, Poland; markoz@cm.umk.pl; 4Centre for Languages & International Education, University College London, 26 Bedford Way, London WC1H 0AP, UK; a.kosinska@ucl.ac.uk; 5Department of Neurology, Faculty of Medicine, Collegium Medicum in Bydgoszcz, Nicolaus Copernicus University in Toruń, Marii Skłodowskiej Curie 9, 85-094 Bydgoszcz, Poland; ela4208@wp.pl; 6Statistical Analysis Centre, Nicolaus Copernicus University in Toruń, Chopin 12/18, 87-100 Toruń, Poland; jwojtasik@umk.pl; 7Centre of Medical Simulations, Faculty of Medicine, Medical University of Gdańsk, Dębowa 25, 80-204 Gdańsk, Poland; gkozera@gumed.edu.pl

**Keywords:** ischemic stroke, cerebral thrombolysis, CCL5 (RANTES), acute, excitotoxicity

## Abstract

One of the key response mechanisms to brain damage, that results in neurological symptoms, is the inflammatory response. It triggers processes that exacerbate neurological damage and create the right environment for the subsequent repair of damaged tissues. RANTES (Regulated upon Activation, Normal T Cell Expressed and Presumably Secreted) chemokine(C-C motif) ligand 5 (CCL5) is one of the chemokines that may have a dual role in stroke progression involving aggravating neuronal damage and playing an important role in angiogenesis and endothelial repair. This study concerned patients with ischemic stroke (AIS), whose CCL5 concentration was measured at various time intervals and was compared with the control group. In addition, the effect of this biomarker on neurological severity and functional prognosis was investigated. Compared to healthy patients, a higher concentration of this chemokine was demonstrated in less than 4.5 h, 24 h and on the seventh day. Differences in CCL5 levels were found to be dependent on the degree of disability and functional status assessed according to neurological scales (modified Rankin Scale, National Institutes of Health Stroke Scale). In addition, differences between various subtypes of stroke were demonstrated, and an increase in CCL5 concentration was proven to be a negative predictor of mortality in patients with AIS. The deleterious effect of CCL5 in the acute phase of stroke and the positive correlation between the tested biomarkers of inflammation were also confirmed.

## 1. Introduction

Stroke is an important health problem affecting patients over 40. It could be caused by vascular occlusion or localized intracranial hematoma. Ischemic stroke is most often triggered by cardiogenic embolism, cerebral impairment or atherosclerosis of the extracranial and intracranial arteries, or a coagulation disorder [1,2,3]. On the other hand, haemorrhagic stroke may be caused by cerebral cavernous malformation (CCM), which belongs to the vascular anomalies of the brain. CCM occurs mainly in the central nervous system (CNS) and is associated with mutations of the CCM genes CCM1/KRIT1, CCM2/MGC4607 or CCM3/PDCD10 [4,5].

The inflammatory response is one of the most important response mechanisms to brain damage that results in neurological symptoms. This initiates several processes that exacerbate neurological damage and creates the right environment for the subsequent repair of damaged tissues [6]. Both the innate and adaptive immune systems are rapidly activated in response to cerebral ischemia and reperfusion injury, leading to the infiltration of various immune cells into the brain parenchyma [7,8]. As a result, leukocytes, regulated by inflammatory mediators, cytokines, chemokines, adhesion molecules and matrix metalloproteinases, flow to the site of the damage [9,10]. Recent evidence points to the importance of these immune responses in the pathogenesis of ischemic brain injury [10,11]. Therefore, immunomodulation (immunotherapy) could be a promising and novel treatment for stroke [12]. Due to the lack of significant data on neural repair in adult brain injuries, such as stroke or traumatic brain injury (TBI), no medical therapies have been developed to support recovery after stroke. The clinically used CCR5 (C-C chemokine receptor type 5) receptor antagonist approved by the FDA in AIDS therapy has only been shown to promote functional recovery in stroke and TBI [12]. Therefore, a study related to the CCL5 chemokine (RANTES) released by T lymphocytes, platelets, endothelial cells, smooth muscle and glial cells was undertaken [13,14,15].

CCL5 is a member of the CC chemokine family participating in the regulation of the pro-inflammatory response by modulating immune cells in tissues [16] and contributes to the pathogenic process of arterial damage and atherosclerosis [17,18], which in turn may be a precursor to the secondary ischemic damage [19,20]. Moreover, CCR5 may be expressed in non-immune cells, especially in astrocytes, microglia and neurons, which are involved in neuronal survival and differentiation [21]. CCL5 and its ligands can therefore play a dual role in stroke progression. On one hand, it could be responsible for aggravating neuronal damage [22,23] and, on the other hand, it may play an important role in the process of angiogenesis [24,25] and endothelial repair [26,27] (Figure 1).

The levels of CCL5, however, in the serum of patients with ischemic stroke, arouses controversy. Several studies reported an increase in the concentration of this biomarker, which was statistically significant compared to controls [22,28,29], while others found no differences or even observed a decrease in the levels of CCL5 [9,30]. For this reason, we decided to assess the level of RANTES in patients with AIS (ischemic stroke) undergoing intravenous thrombolysis to determine its role in the aetiology of stroke.

In addition, we were unable to find published information on CCL5 levels in different stroke subtypes as well as its impact on patient mortality. There is little information evaluating CCL5’s role in predicting the functional outcome of people with AIS. Moreover, the lack of unequivocal evidence of the neuroprotective role of CCL5 and its impact on the clinical effectiveness of post-stroke patients provides grounds for determining its levels at different stages of stroke. Studying the concentration of this chemokine in the inflammatory response will also provide a further understanding of the mechanism of stroke. Additionally, this study could help in the development of new pharmacological strategies. This may be an important target for therapeutic modifications. Since biomarker predictors, which could effectively forecast the clinical outcomes of patients with stroke, have been sought in recent years, it seems interesting and beneficial to determine their function in clinical aetiology.

Therefore, we decided to investigate the role of CCL5 through extended clinical follow ups of stroke patients by measuring the CCL5 concentration at different time intervals, <4.5 h, 24 h and the seventh day, and compare them with the control group. The effect of this biomarker on neurological severity and functional prognosis was also studied.

## 2. Materials and Methods

### 2.1. Study Group

The study of the CCL5 chemokine (RANTES) involved assessing 125 patients with acute ischemic stroke, who underwent thrombolysis treatment at the Department of Neurology of University Hospital in Bydgoszcz, with an average age of 64 years. Patients of the Neurology Clinic were selected diagnostically by excluding patients with cardiovascular diseases, immunological disorders, cancer, kidney diseases or injuries from the study. This research was given the consent of the Bioethics Committee No. KB 637/2016 and patients participated voluntarily in this research. In this study group, stroke was diagnosed based on ICD 10 criteria and confirmed by the clinical assessment and neuroimaging using computed tomography (CT) or magnetic resonance imaging (MRI). Different types of strokes were diagnosed based on the currently used TOAST classification of ischemic stroke. The assessment included atherosclerotic changes in the large intracerebral and small cerebral vessels and cardiovascular embolism. All diseases accompanying the selected group of patients were diagnosed in accordance with the applicable standards.

The control group consisted of 28 people aged 27 to 72 who were assessed as healthy individuals based on the test results and clinical interviews. Their participation in this study was voluntary.

### 2.2. Biochemical Testing

On admission to the Neurology Department of the University Hospital, in addition to routine diagnostic tests, blood samples were also collected from the basilic vein into tubes with cloth activator and gel separator. Samples of this blood were used for the determination of the chemokine CCL5, which was measured at various time intervals. After obtaining the biological material, the blood was centrifuged (3000× *g* for 15 min) and then poured into Eppendorf tubes. The biospecimens were stored at −80 °C until biochemical analysis was performed. The CCL5 chemokine was determined in serum samples from patients with acute ischemic stroke within <4.5 h, within 24 h and after 96 h. For comparison, the concentration of RANTES in the serum of control subjects was determined. According to the manufacturer’s instructions, all samples were marked by ELISA, i.e., enzyme-linked immunosorbent assay, using a commercial RANTES ELISA kit for humans (Cloud Clone, Wuhan, China) with a sensitivity of 0.059 ng mL^−1^.

### 2.3. Statistical Methods

The R programming language (version 4.1.2) was used for statistical analyses. Data were tested for normal distribution and equality of variance.

The Shapiro–Wilk test of normality was used for the analysis, the Mann–Whitney U test for median comparison, the chi-square test for binary variables, and the Spearman rank test for correlation assessments. *p* < 0.05 was considered statistically significant. In the statistical analyses, ROC curves were also employed, and the optimal cut-off points of the biomarker were determined.

## 3. Results

All biological material was collected from 125 patients at the time they were admitted to the Neurology Department of the University Hospital in Bydgoszcz. The size of the study group at the time of discharge was 121 patients, 113 patients after 3 months and 96 patients after a year post-discharge. Based on the modified Rankin Scale (mRS), patients were divided into two subgroups, with a favourable (0–2 points) and unfavourable (3–6 points) functional score, and based on the scores, they were qualified for intravenous thrombolysis. A favourable mRS score was observed for 26 patients (21%) at the time of hospital admission, 91 patients (75%) at the time of discharge, 85 patients (75%) after 3 months and 79 patients (82%) after 1 year since stroke. During the period of this study, 15 patients died, including 8 within the first 3 months since the stroke. The cause of death was due to neurological disorders—cerebral and extracerebral. On hospital admission, coronary artery disease was observed in 27 (21.6%), diabetes in 52 (41.6%), atrial fibrillation in 15 (12.0%), gout in 6 (4.8%) and renal failure in 3 (2.4%) patients. Before an ischemic event, 3 (2.4%) patients were treated with anticoagulants and 35 (28%) with statins. Patients with a favourable Rankin Score were younger (*p* = 0.002), and had lower scores at discharge, 3 months and 1 year after discharge (*p* < 0.001, *p* < 0.001 and *p* = 0.003, respectively). This group of patients had fewer NIHSS (National Institutes of Health Stroke Scale) neurological changes at the time of admission and discharge (*p* < 0.001 and *p* < 0.001). They were less likely to have an infection (*p* = 0.007) and coronary artery disease (*p* = 0.028). They also had lower systolic blood pressure (*p* = 0.003) and were less likely to take antibiotics (*p* = 0.007). Clinical characteristics of subgroups of thrombolytic stroke patients with favourable and unfavourable mRS outcomes at the time of admission are presented in Table 1.

The level of patients’ disability was assessed according to the mRS at the time of hospital admission, discharge, 3 months and 1 year after the stroke and according to the NIHSS at the time of admission and discharge. Patients were divided into two subgroups with favourable (mRS: 0–2 points; NIHSS: ≤3 points) and unfavourable (mRS: 3–6 points; NIHSS: >3 points) outcomes. It was found that serum CCL5 concentrations in the subgroup of patients with a favourable functional score were lower than those with an unfavourable functional score as shown in Table 2 and Table 3. There was a statistically significant difference between the median CCL5 values determined at the time of admission (<4.5 h) and assessed according to the mRS scale one year after the stroke (94.2 ng mL^−1^ with IQR 9.9 vs. 72.4 ng mL^−1^ with IQR 29.3, respectively, *p* = 0.011) and between CCL5 on day 7 and NIHSS at admission, Table 3 (65.4 ng mL^−1^ with IQR 21.5 vs. 47.3 ng mL^−1^ with IQR 17.7, respectively, *p* = 0.032).

Statistically important differences between the CCL5 concentration values in the group of stroke patients compared to the control group were also demonstrated. The median CCL5 concentrations were 75.9, 68.0 and 57.8 ng mL^−1^ vs. 45.9 ng mL^−1^, respectively, with *p* < 0.001, *p* < 0.001 and *p* = 0.005, as shown in Table 4.

There were no statistically significant correlations observed between the level of CCL5 and the values of the mRS and NIHSS scales, and no effect of statins on the level of the biomarker was found.

Receiver Operating Characteristic (ROC) curves were prepared to assess the effect of CCL5 on the level of functional independence. Since the CCL5 levels were assessed at the time of admission to the hospital (<4.5 h), they considerably distinguished patients with favourable and unfavourable functional results for these concentrations, hence additional analyses were performed. The Areas Under the ROC curves (AUCs) were 0.637, 0.628, 0.631, and 0.799, respectively, for the favourable mRS functional scores at the time of admission, discharge, after 3 months, and after 1 year since the stroke. The cut-off points for these curves are, respectively, 66.53 ng mL^−1^ (sensitivity 55.0%, specificity 75.4%), 75.90 ng mL^−1^ (sensitivity 55.9%, specificity 70.0%), 83.85 ng mL^−1^ (sensitivity 65.5%, specificity 64.7%) and 87.20 ng mL^−1^ (sensitivity 72.2%, specificity 85.7%). The summary of the results is illustrated in Figure 1.

CCL5 concentrations were compared for different stroke subtypes: lacunar cerebral infarcts (LACI), posterior circulation infarcts (POCI) and partial anterior circulation infarcts (PACI). The lowest levels of this biomarker are found in patients with the LACI subtype. Statistically significant differences between the values of this biomarker in patients with LACI stroke and POCI were observed in <4.5 h (69.6 ng mL^−1^ with IQR 24.6 vs. 90.67 ng mL^−1^ with IQR 19.6, *p* = 0.018). The obtained results are presented in Table 5 and Figure 2.

The effect of CCL5 concentration on the likelihood of patients dying was also assessed (Figure 3). At concentrations above 75 ng mL^−1^, a sharp increase in stroke mortality was observed. A similar relationship was also demonstrated during onset (value between 50 and 100 ng mL^−1^) and on the first day of stroke (>60 ng mL^−1^ and 100 ng mL^−1^).

In our earlier studies, we assessed the participation of IL-6 and TNF-α [31,32] in AIS and therefore we examined the correlation between the parameters of inflammation (IL-6, TNF-α, CCL5) as shown in Figure 4. Interestingly, we found a statistically significant correlation between the TNF-α concentration assessed on the seventh day and the CCL5 concentration assessed after 24 h (R = 0.81). In addition, weaker (*p* < 0.1) positive correlations were also observed between the concentration of the tested biomarker determined at onset and after 24 h and TNF-α assessed <4.5 h and after 24 h (R = 0.29; 0.14; 0.54, respectively). In addition, we also found such a relationship with IL-6 assessed in less than 4.5 h and on the seventh day, and CCL5 in less than 4.5 h (R = 0.16; 0.22) since the stroke.

We also studied the effect of clinical parameters on the CCL5 concentration. We found no statistically significant differences in RANTES levels in stroke patients with diseases such as diabetes, atrial fibrillation and hypertension. In patients with ischemic heart disease *(p* = 0.036) and smokers (*p* = 0.046), we only observed differences in the population of people with favourable and unfavourable mRS results (Table 6).

## 4. Discussion

In this study, higher CCL5 concentrations have been observed at 4.5 h, 24 h and on the 7th day after stroke compared to the healthy patients (*p* < 0.001, 0.001 and 0.005, respectively, Table 4). Unfortunately, there are various reports [9,22,28,29,30,33] with conflicting data as to whether circulating CCL5 levels rise or fall after an ischemic stroke. Some studies confirmed our observations, where elevated levels of this biomarker were detected in both blood and cerebrospinal fluid [22,28,29], while others showed no differences in CCL5 levels compared to controls or in patients with ischemic stroke over time [9,16,30]. In turn, R. Badacz et al. showed a reduced level of CCL5 [33].

The discrepancies in the results may have various causes. The sequence and time of leukocyte response depend on the time profile of expression of inflammatory mediators [31,32,34]. Moreover, in some of these studies [28,33], there was no control group to establish levels of CCL5 in healthy patients. Differences could have also resulted from methods of collecting biological material including time of collection. Gender, age, ethnicity, associated diseases, extent and time of reperfusion and type or evolution of stroke could also explain inconsistencies observed in different studies [34,35,36,37].

In our research, variations were seen between the concentration of CCL5 at different time intervals. However, the highest levels of CCL5 were found within the first hours after the appearance of clinical symptoms in stroke patients. Probably, a higher level of RANTES in plasma may indicate more active inflammation, which is confirmed by other authors [38]. This has also been confirmed by our previous studies, in which we proved that the overexpression of inflammation and oxidative stress occur shortly after a stroke [31,32,39]. The aggravation of this condition in the brain is mediated by leukocytes. Their infiltration is initiated within a few hours of stroke by neutrophils and then within 24 h by monocytes and lymphocytes [30].

In addition, we attempted to determine the concentration of RANTES in various subtypes of ischemic stroke, LACI, POCI and PACI (Table 5, Figure 2). We have shown that the lowest concentrations of this biomarker occur in patients with the best prognosis subtype of LACI, i.e., lacunar infarct. Statistically significant differences in CCL5 concentrations were reported in the LACI and POCI subtypes and statistically insignificant differences were seen in the LACI and PACI subtypes due to the insufficient number of patients. It seems that CCL5 may be a good parameter differentiating the types and subtypes of stroke, which is consistent with the literature sources, especially related to haemorrhagic and ischemic stroke as well as ischemic lacunar and cardiogenic stroke [16].

This study’s results may indicate that the concentration of this parameter increases with the severity of the AIS stroke, which translates into a worse prognosis (Figure 2). On the seventh day after the stroke, a decrease in the concentration of this parameter was observed, which further confirms our hypothesis. The number of people with favourable neurological outcomes is increasing over time as illustrated by Table 2 and Table 3.

Animal studies support the assumption that RANTES may be actively involved in stroke initiation and progression. For example, CCL5 knockout mice showed a reduced volume of the ischemic area, significantly reduced blood–brain barrier permeability, and reduced leukocyte adhesion and platelet adhesion [40]. Injection of a CCL5 inhibitor into tested animals resulted in both a reduction in infarct size and an improvement in neurological outcomes after stroke [41]. Deletion of the CCR5 receptor gene has a protective effect against cerebral ischemia and reperfusion injury [12,16,42]. In addition, Q. Kong et al. [29] proved that the plasma concentration of RANTES is positively correlated with the atherosclerosis of the cardiovascular and cerebral arteries in patients with ischemic cerebrovascular disease. This confirms the view that RANTES may actively participate in the formation of atherosclerotic plaques and their progression [38].

Equally, the increase in CCL5 concentration after ischemic stroke may protect neurons by producing neurotrophic factors in peri-infarct areas [22]. CCL5 affects vasodilation, inhibits platelet aggregation, and induces angiogenesis. The neuroprotective role is confirmed by some researchers, such as H. Tokami et al. [22] observed an increased plasma RANTES concentration in patients with ischemic stroke on day 0 from the onset of infarction symptoms. This concentration was statistically significantly higher than in the control group and strongly correlated with the concentration of the BDNF (brain-derived neurotrophic factor), EGF (epidermal growth factor) and VEGF (vascular endothelial growth factor), which may confirm the important role of CCL5 in endothelial repair.

F. J. Julián-Villaverde et al. observed a decrease in CCL5 levels in patients with haemorrhagic stroke compared to patients with acute ischemic stroke. Based on this, they concluded that CCL5 may act as a neuroprotective factor in ischemic stroke and that a lower level of CCL5 on admission increases the stroke volume and worsens the patient’s prognosis [16].

The potential neuroprotective role of RANTES has also been confirmed by studies conducted on animal models. CCR5-deficient mice undergoing ischemic stroke had a larger infarct size with increased neuronal death and neutrophil infiltration compared to wild-type, control animals [21,43].

These conflicting literature reports support the hypothesis of a dual role of CCL5. This chemokine may play a pro-inflammatory, potentially harmful role in stroke progression and a neuroprotective, i.e., protective, role in relation to nerve cells.

As a result of this, extended follow ups of patients with ischemic stroke were carried out, which involved not only measuring their circulating CCL5 levels at various times after the onset of stroke but also examining whether these values could predict the neurological severity and functional prognosis.

The correlation between the neurological scales (mRS and NIHSS) and the biomarker was examined but a meaningful connection was not found at the time points studied. Similarly, T. Garcıa-Berrocoso et al. [9] and Zaremba et al. [30] found no relationship between the level of RANTES at the time of hospital admission, after 24 h and on the 7th day, and the NIHSS and SSS (Scandinavian Stroke Scale) neurological scales as well as the degree of disability measured by BI. They were only able to show such an association between CCL5 determined on the 3rd day of stroke and the NIHSS scale assessed at hospital admission [9].

In addition, we also analysed the results for the m Rankin Scale, divided into two categories that were correlated to good (mRS ≤ 2) or poor (mRS > 2) prognosis at hospital admission, discharge, and 3 months and 1 year after stroke. There were no differences between the prognostic groups in patients with AIS at admission, discharge and after 3 months, which agreed with the latest reports by F. J. Julián-Villaverde et al. [16]. As the only ones, we were able to demonstrate such a relationship only after a year in the onset time (Table 2).

By assessing the CCL5 level with the NIHSS scale (NIHSS ≤ 3 or >3) at the hospital admission and discharge, we confirmed [15] that in the group of patients with ischemic stroke with a good prognosis, CCL5 levels were lower, and on the seventh day even with *p* = 0.032 (Table 3). These research results may indicate its potential predictive role.

So far, we are the only ones who showed a relationship between the concentration of CCL5 and mortality due to AIS (Figure 3). We also performed an in-depth analysis of the work by R. Badacz et al. Their research proved that higher mortality in AIS patients was associated with lower CCL5 levels due to cardiovascular complications. However, the authors did not compare the CCL5 concentration with the control group of healthy people. Moreover, the study group included patients with transient ischemic attack (TIA) [33]. Additionally, the study group with AIS was divided only into coronary and carotid ischemic events (MACCE) and non-MACCE. The authors reported that serum RANTES concentrations below 45.5 ng mL^−1^ were linked to an increased risk of MACCE.

Our research, however, cannot be related to the above study due to a different selection of the study group. We have found that the mortality rate of patients with AIS increases with the increase in CCL5 concentration. In addition, the cut-off point in onset time, after 24 h and 7 days, has been determined between 75 and 100 ng mL^−1^, which has not been reported before. Other authors have observed that an elevated CCL5 value in AIS patients may predict three-month mortality and poor patient prognosis and that it may act as a negative predictor of clinical effectiveness [28,29].

Higher plasma levels of RANTES indicate more active inflammation and fibroproliferation in atherosclerotic plaques [29]. This also supports the view that RANTES contributes to the intensive development of AIS. However, there is no clear information about its role in the treatment of stroke. CCL5 is involved in immune s surveillance, inflammatory response, tumour formation and metastasis, and in the pathogenesis of inflammatory diseases in asthma and cancer. In addition to its direct involvement in the mediation of immune processes, CCL5 acts as a suppressor of cognitive functions and synaptic connections in the brain [44]. CCL5, despite its pleiotropic effects, is of interest to many researchers seeking new therapies focusing on neuronal repair in stroke and traumatic brain injury. Selective activation and/or inhibition of receptors at a specific time and concentration may be the key factor in designing therapeutic drugs used in the treatment of stroke. It has been shown that people with the CCR5-Δ32 mutations have a more promising outcome after a stroke, increased motor regeneration, and reduced cognitive deficits [12]. So far, scientists have only been able to conclude that CCL5 inhibitors are a promising class of antiviral drugs, e.g., for HIV. It has been proven that the loss of function of the CCR5 gene modulates the risk of HIV transmission and prevents the pathogenesis of HIV infection [44].

The confirmation of the harmful effect of CCL5 in the acute phase of stroke is supported by the correlation observed between the biomarkers of inflammation, IL-6 and TNF-α [31,32], previously determined by our research centre, which is among the first to be released during ischemia in AIS (Figure 4). This relationship can, especially, be observed between the concentration of CCL5 determined during 24 h and TNF-α assessed on the seventh day. This may indicate a modulating effect of CCL5 on the immune system. However, there is still little information describing such relationships. In the work of R. Badacz et al., only a negative correlation between the concentration of CCL5 and IL-6 was found [33].

The results of our study may indicate that this overexpression of cytokines may have a negative impact on the long-term prognosis of the patient and inflammatory biomarkers can be employed as predictors. On the other hand, it should be noted that the severity of inflammation is the result of the body’s reaction to hypoxia. In the dynamically developing process of inflammation, CCL5, with the participation of other inflammatory mediators, performs a repair function by spreading at the site of damage to support tissue regeneration. It was found that CCL5 differentiates macrophages from the M1 phenotype, which exhibits bactericidal properties and secretes pro-inflammatory cytokines in response to inflammation, involved in wound healing. In addition, CCL5 limits the formation of foam cells from macrophages even though it may also be a key factor in the progression of atherosclerotic disease [14].

In addition, an attempt was made to compare CCL5 concentrations in the group of men and women similarly to the work of F. J. Julián-Villaverde et al. [16], which indicated a decrease in the CCL5 level in women with AIS in comparison to men. Moreover, it was observed that ischemic stroke mainly affected men. However, to accurately track and unambiguously assess their prognosis, it would be necessary to conduct studies on a larger group in parallel with other medical centres.

We also studied the effect of clinical parameters on CCL5 concentration. We found no statistically significant differences in RANTES levels in stroke patients with diseases such as diabetes and hypertension. However, there are reports describing the contribution of these clinical factors to the concentration of pro-inflammatory cytokines [45,46]. In patients with ischemic heart disease *(p* = 0.036) and smokers (*p* = 0.046), we only observed differences in the population of people with favourable and unfavourable mRS results (Table 6). Despite the efforts to study as a unified population as possible, practically all patients had accompanying diseases. Therefore, it is difficult to determine their direct contribution to CCL5. However, the conducted study has some limitations. The need to express informed consent for additional blood sampling for CCL5 determination resulted in the exclusion of patients with impaired consciousness or aphasia. For this reason, a selected group of patients was characterized by mild and moderate neurological deficits. The presence of comorbidities that accompany stroke patients should also be considered, i.e., hypertension, diabetes, atrial fibrillation, and others.

The above limitations may affect the interpretation of the study results and therefore should be continued in a larger population as a multicentre study.

## 5. Conclusions

The study of biological mechanisms of RANTES involved in the inflammatory response will provide a thorough understanding of the pathology of stroke and, consequently, may help to develop new pharmacological strategies. Measurement of this biomarker at various times after an ischemic event may be important in predicting the outcome of disability levels or functional prognosis. Elevated CCL5 content in the blood of people with AIS may be a negative predictor of clinical effectiveness and even an indicator of mortality among stroke patients. However, clinical trials on a larger population of people with AIS, and across different medical sites, should be performed to confirm the pro-inflammatory role of RANTES in both the acute phase and the progression of stroke. One should also consider its neuroprotective effect by examining its levels in a wider time window.

The study of biochemical markers is an important, new direction in the diagnosis of stroke. The development of diagnostic tests requires the determination of those biomarkers’ acceptable norms depending on gender, age, lifestyle, drugs or genetic traits. Designing a panel for sequencing genes responsible for cerebral vascular malformation may also be a diagnostic standard [4,5]. A rapid diagnostic test can be particularly useful in emergency medical care that would allow for fast decisions regarding further treatment of the patient.

## Data Availability

Data are available from the authors.

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
