# Peer review of "The Potential Role of RANTES in Post-Stroke Therapy"

_cells, 2023, doi:10.3390/cells12182217_

Round 1

Reviewer 1 Report

The authors addressed all suggested points.

The English requires only minor adjustments.

Author Response

Dear Reviewer,

thank you for your in-depth manuscript analysis and suggestions, which significantly helped us to improve our manuscript. Answers in the attachment.

Yours faithfully,

Hanna Pawluk

Reviewer 2 Report

I am fine with the revision. appart from the point with the presentation of the interquatile range (IQR) versus Q1, Q3. 

Infact the presentation is correct, but unusual. If there is Ok for the editor, I will go with it. Otherwise I will replace Q2, Q3 with IQR.

Appart from that the paper is ready.

Author Response

(The authors gave the same response as above.)
